# Etiology of severe mastitis in French dairy herds

Olivier Salat[1☯¤a]*, Guillaume Lemaire[1☯], Luc Durel[2], Florent Perrot[1☯¤b]

1 DVM, Clinique Vétérinaire de Haute Auvergne, ZAC Montplain Allauzier, Saint-Flour, France, 2 DVM, Global Technical and Medical Information Manager, VIRBAC, Carros, France

☯ These authors contributed equally to this work.
¤a Current address: Diplomate of the European College of Bovine Herd Management (ECBHM), Berlin, Germany
¤b Current address: Resident of the ECBHM, Berlin, Germany
* olivier.salat@free.fr

**Data Availability Statement:** All relevant data are within the paper and its supporting information files.

## Abstract

Severe clinical mastitis is not so frequent in dairy cows, but it can have a significant economic impact due to its high mortality rate and adverse effects on milk production. Surveys about the cause of mastitis typically provide overall findings without connecting them to a specific medical condition. There are few studies on the specific etiology of severe mastitis. Only etiological results obtained during field studies are available in France, and the number of mastitic milk samples analyzed is always low. In recent years, veterinary clinics have improved their ability to identify bacteria causing bovine mastitis using a widely used method. This in-clinic milk culture made it possible to better understand the etiology of mastitis according to the symptoms observed. Regarding clinical mastitis, veterinarians treat severe cases differently than less severe ones. Based on data from nineteen veterinary clinics in France and over 2000 severe mastitis cases, the current study revealed that Enterobacteriaceae, particularly *Escherichia coli*, is the most common cause, isolated on average from 53.9% of milk samples. This information is highly reliable for practitioners to quickly and effectively treat the condition, because early and targeted treatment is crucial to avoid the complication of endotoxic shock.

## Introduction

Clinical mastitis (CM) in dairy cattle remains the most expensive production disease and the primary reason for antimicrobial use. The *International Dairy Federation* classifies clinical mastitis into three categories [1]:

- mild CM when milk is the only modification,

- moderate CM when the udder is inflamed,

- severe CM with systemic signs, hyperthermia, decreased ruminal motility, less or more anorexia, and depression.

**Funding:** the author received no specific funding for this work

**Competing interests:** the authors have declared that no competing interests exist

**Abbreviations:** CM, clinical mastitis; SA, *Staphylococcus aureus*; EC, *Escherichia coli*; SU, *Streptococcus uberis*; NaS, non-aureus Staphylococci.

The distribution between these three types of CM is the following: 48–55% of mild, 36–45% of moderate, and 9–21% of severe CM [2–4].

CM can have severe consequences for cows, including pain, reduced milk production, culling, and even death. However, the financial losses associated with this condition are often not accurately calculated, particularly for mastitis caused by Gram-negative (Gram-) bacteria. These cases can lead to substantial financial losses of up to 450/case [5,6] due to a significant reduction in milk yield [7]. Since Gram- bacteria are more commonly found in severe CM [8], the economic consequences are likely even higher in these cases. Severe CM can cause a sudden and lasting decline in milk production, agalactia, anticipated culling and mortality [9]. In Ireland, *E. coli* (EC) mastitis is the primary cause of death among dairy cows [10]. Consequently, prompt antibiotic treatment is crucial, and waiting for the results of milk cultures is not an option, generally.

Severe CM can progress rapidly and lead to systemic inflammatory shock. Severe cases (with generalized signs, such as fever, decreased ruminal motility, anorexia, distress) are medical emergencies and should immediately be treated using protocols developed with veterinary input [11]. It is crucial to treat this kind of CM with antibiotics promptly for two reasons. First, the severity of CM is related to the amount of bacteria in the udder [12], and second, the risk of bacteremia increases as the severity of CM worsens [13,14]. Research has shown that extralabel use of systemic antibiotics (such as injectable ceftiofur) is beneficial for the treatment of septicemia that occurs in many cows affected with severe CM [15,16]. Moreover, in France as in other areas in the world, several fluoroquinolones are registered for use in dairy cows. When injected parenterally, these molecules reach effective concentrations, not only in the blood to stop sepsis but also in the udder to eliminate infection [17]. They are among the few antibiotics registered for EC severe CM. Knowing the dominant pathogen causing the infection can help determine the appropriate antibiotic for immediate treatment. Use of fluoroquinolones is authorized in France, but only on the basis of a milk culture test and an antibiotic sensitivity test. In-clinic milk culture has developed for the past 20 years [18], improving the learning of bovine CM among veterinary practitioners. Rather than epidemiological knowledge based on global bacteriological results from all kinds of mastitis, bacteriological results related to clinical appearance are more relevant and valuable for veterinary practitioners. EC-mastitis has long been a synonym of severe CM; however, other pathogens, such as *Streptococcus uberis* (SU), have recently gained attention [19]. This study aims to explore the French situation regarding severe CM.

## Material and methods

Nineteen veterinary clinics from the leading French dairy areas participated in this study, after a call to all practitioners who have been trained in the bacteriological analysis method used. These clinics might record their bacteriological results by severity level (mild, moderate or severe clinical mastitis) and share the same bacteriological analysis method. Veterinarians classified their milk test results by the type of CM sampled. According to local denominations, this study included CM cases named "severe mastitis," grade-3 mastitis, or "acute mastitis with general symptoms". Analyze records mainly concern year 2022, but some clinics have also sent data from 2021 and 2020. All the veterinary clinics that took part in this study agreed to share their data in order to provide a nationwide etiological profile of severe mastitis (no other permit required).

On the dairy producer's call, local veterinarians, and occasionally farm staff, took milk samples; the staff had received prior training to collect milk aseptically. Briefly, teats were cleaned with a damp cloth and soap, rinsed and dried with paper. The teat end was disinfected with

alcohol-impregnated gauze. The first jets were removed and one or 2 following jets directed into the pot held as horizontally as possible. Milk samples were taken as soon as mastitis was detected by the farmers, and as soon as the sick cow has been seen by the veterinarian. As these cases were considered emergencies, the time between call and veterinary intervention was usually less than 2 hours. Veterinarians or the farm staff assessed clinical severity using a dedicated clinical report form (S1 File) after clinical examination. Only the bacteriological data of clinically severe cases were considered for the analysis.

Veterinary clinics enrolled in this study performed standardized microbiological diagnostic tests as per the inspirations of the *National Mastitis Council* [20]. Briefly, thirty microliters (30μL) of each well-mixed milk sample were plated (30μL per single plate) onto a series of media with a sterile calibrated loop: 1) 5% sheep blood agar (COS—Biomérieux, Lyon, France) to validate sample quality, 2) 5% sheep blood agar added with 15 mg/L nalidixic acid and 10 mg/L colistin sulfate (CNA—Biomérieux), to identify Gram-positive bacteria and 3) Hektoën enteric agar (HEKT—Biomérieux) to allow the detection of Enterobacteriaceae. Plates were incubated at 37°C under aerobic conditions and read at 12, 24, and 48 h. Organisms were enumerated and identified using laboratory techniques described elsewhere [20]. Catalase reactivity (3% $H_2O_2$) was performed on colonies grown on CNA (CNA+) (Fig 1). Colonies being both CNA+, catalase+, and displaying a double-hemolysis, or coagulase+ colonies were identified as *Staphylococcus aureus* (SA). Gram staining was performed on the other CNA+ and catalase + colonies to differentiate non-aureus staphylococci (NaS) from *Bacillus* spp, *Corynebacterium* spp, yeast, *Prototheca* spp, or fungi. For Gram+ and catalase- colonies, aesculin hydrolysis was tested. The Lancefield test was performed in case of a lack of reaction after two hours of incubation. If a tested colony was aesculin+, then the colony was plated onto Bile-aesculin agar to differentiate SU from *Enterococcus* spp.. *Trueperella pyogenes* was recognized as small colonies growing slowly on CNA agar, with late β hemolysis, catalase-, aesculin- and pleomorphic or coccoid rods, with pairs often V-shaped when Gram-stained. For HKT agar-grown organisms, colonies were cultivated onto CPSO agar and into TSI (triple sugar iron) broth (Biomérieux), and colony color and growth and medium color allowed their identification. We only differentiated EC from other Gram- species to prevent mistakes. EC appeared as yellow colonies on a salmon background on the HEKT media, hydrolyzed glucose and lactose in the TSI broth, and grew as pink colonies on CPSO agar. Samples that yielded two different bacterial species were grouped as "mixed culture," whereas samples yielding three or more distinct species (at least one CFU per species) were considered contaminated. A minimum of ≥1 CFU/10μL of milk plated on milk culture was required to assess the presence of major mammary pathogens (SA, *Streptococcus agalactiae*, *Streptococcus. dysgalactiae* and SU, and coliforms) and *Enterococcus* spp. [21]; for all other pathogens the cut-off was 2 CFU/10μl.

## Results

A total of 2087 severe CM cases were examined across nineteen veterinary clinics, but the number of cases varied greatly per clinic, ranging from 15 to 464 (S2 File). To prevent bias in the data, the number of records was randomly limited to 100 per clinic for those with high numbers. Corrected prevalences were calculated for these clinics based on the 100 selected files. Conversely, all records were kept for clinics with less than 100 results, and their prevalence data were not adjusted. The final analysis included 1210 results, with Table 1 showing the actual and corrected distribution of the isolated pathogens. The raw data is provided as supplemental material, and only a few contaminated samples were excluded from the analysis.

Based on bacteriological analyses conducted by the participating clinics, the percentage of Gram- bacteria ranged from 42% to 81%, averaging 59.7%. Only three clinics (C, H, and Q)

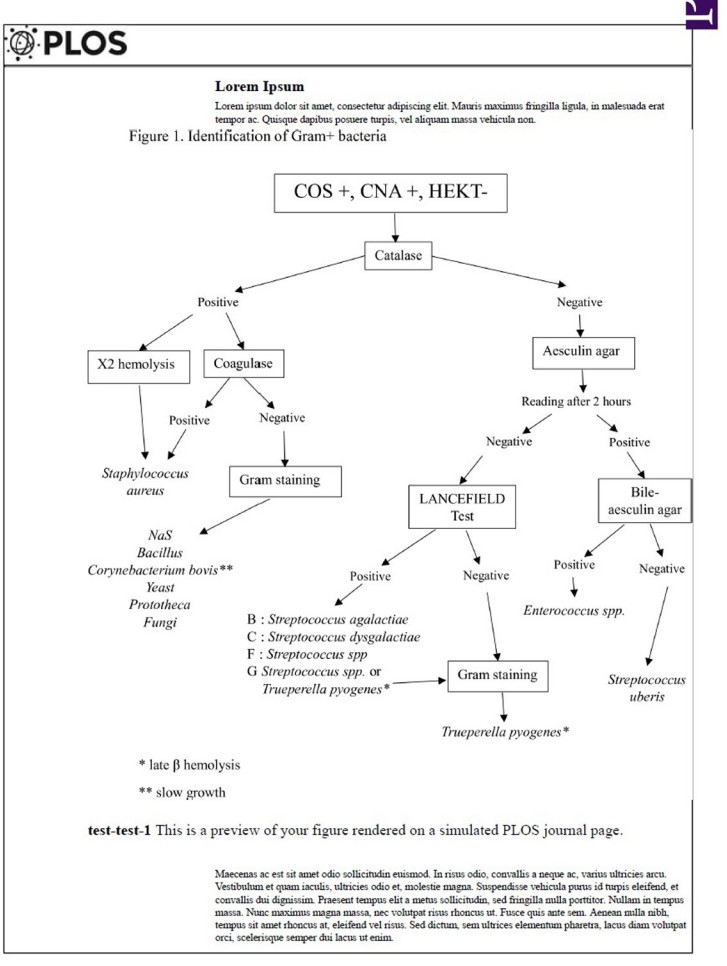

**Fig 1. Summary diagram for identifying Gram-positive bacteria.**

**Table 1. Raw (2087 data) and corrected (1208 data) distribution of pathogens isolated from severe clinical mastitis.**

|  | Raw distribution | | Corrected distribution | |
|---|---|---|---|---|
| Isolated pathogens | Nb of isolates | % | Nb of isolates | % |
| *Escherichia coli* | 1118 | 53.6% | 651 | 53.9% |
| other Gram- | 128 | 6.1% | 90 | 7.5% |
| *Staphylococcus aureus* | 182 | 8.7% | 97 | 8.0% |
| NaS[a] | 132 | 6.3% | 75 | 6.2% |
| *Streptococcus uberis* | 239 | 11.5% | 131 | 10.8% |
| *Streptococcus dysgalactiae* | 46 | 2.2% | 20 | 1.7% |
| other S*treptococci/Enterococcus* | 52 | 2.5% | 41 | 3.4% |
| *Trueperella pyogenes* | 28 | 1.3% | 15 | 1.2% |
| other Gram+ | 5 | 0.2% | 5 | 0.4% |
| No growth | 148 | 7.1% | 79 | 6.5% |
| Yeasts, fungi | 9 | 0.4% | 4 | 0.3% |

[a]Non-aureus staphylococci.

had more Gram+ bacteria. Meanwhile, 15 veterinary clinics had more than half of their bacteriological results showing Gram- bacteria (refer to Table 1). EC was the most commonly isolated pathogen, accounting for 31% to 72% of all isolated bacteria (Table 2). The second most frequently isolated pathogen was SU (in 11/19 veterinary clinics), which accounted for 7% to 20% of the results. SA was the second most often isolated pathogen in 6/19 veterinary clinics.

## Discussion

The relevance of the results of in-house microbiological analysis of milk samples from clinical mastitis has been previously questioned because they reveal a limited set of phenotypic characteristics and do not appropriately differentiate the variety of species that can be present in milk and will grow on blood agar [22]. The bacterial milk culture techniques used in French veterinary clinics can vary in terms of relevance, not to mention the skills of the operators. As a general recommendation in the UK, and in Sweden, the use of in-house and on-farm milk culture must be periodically backed up with quality control using specialist laboratories [23]. However, in our study, all clinics used the same standardized identification technique, which differed from the different methods used by Danish veterinary clinics where the commercial test CHROMAgar™ Orientation (Paris, France) was used alongside other tests for base [22]. According to unpublished data, the proposed identification technique had excellent matches for EC and SA (with Cohen's kappa coefficients of 0.99 and 0.95, respectively), and suitable matches for streptococci species (Cohen's kappa coefficient >0.8) (S3 File). It must be acknowledged that differentiating coliforms other than EC using this technique is less dependable. Due to this issue and their infrequent occurrence, we grouped them and labeled them "other Gram-".

Using phenotypic techniques, it is not easy to differentiate between SU and other Gram+, catalase-, aesculine+, and Bile-aesculin-negative species (such as *Lactococcus* spp., *Aerococcus* spp., and *Helcococcus ovis*). Therefore, the prevalence of SU is overestimated. In addition, it is unclear if the other bacteria mistaken for SU are pathogenic and associated with severe CM. However, any errors in identifying these bacteria have a limited impact on the final epidemiological distribution.

In our study, the Enterobacteriaceae family (Rahn, 1937), mainly represented by EC, accounted for 60% of the bacteriological results and approximatively the two thirds of the isolated pathogens. Thus, more than half of severe mastitis cases involved EC. Similarly, studies in North America have found that Enterobacteriaceae are primarily responsible for severe CM, with their presence ranging from 54% to 75.6% [3,15,24]. While coliforms are the most dominant type of Gram- bacteria, EC is not always the most prevalent, with *Klebsiella* spp. sometimes being just as frequent. However, recent studies in Germany have found that coliforms are not as dominant, only being isolated in 17.1% to 42.2% [2,4,25] of severe CM cases. However, it is essential to note that the number of dairy herds enrolled in these studies was limited, which raises questions about their ability to represent the etiological situation of severe CM in Germany. Research conducted in several European countries has shown that EC is the most common pathogen found in cows with severe mastitis, accounting for 39% of cases in Belgium [26]. Similarly, Enterobacteriaceae were more frequently present than streptococci in severe mastitis cases in Austria [27]. As similarly reported in German studies, Gram+ pathogens caused half of the severe mastitis in a few of our French practices. While EC remained the most commonly isolated pathogen, SU and SA increased. Both Gram+ and Gram- bacteria have been found to possess numerous virulence factors; however, the apparent epidemiological discrepancy between farms needs to be further explored.

**Table 2. Milk culture records by veterinary clinic.**

| VETERINARY CLINIC (home department[a]) | Gram- (%) | Gram+ (%) | no growth (%) | 1rst most often isolated pathogen[b] (prevalence, %) | 2nd most often isolated pathogen[b] (prevalence, %) |
|---|---|---|---|---|---|
| A (43) | 56 | 36 | 8 | *E coli* (51) | *S. aureus* (10) |
| B (50) | 65 | 27 | 8 | *E coli* (53) | *S. aureus* (10) |
| C (85) | 47 | 53 | | *E coli* (47) | *S. aureus* (26) |
| D (76) | 81 | 19 | | *E coli* (71) | *S. aureus* (14) |
| E (63) | 53 | 44 | 3 | *E coli* (35) | *Str. uberis* (12) |
| F (50) | 65 | 25 | 10 | *E coli* (65) | *Str. uberis* (10) |
| G (50) | 66 | 23 | 11 | *E coli* (60) | *Str. uberis* (9) |
| H (63) | 42 | 46 | 12 | *E coli* (35) | *Str. uberis* (18) |
| I (50) | 79 | 19 | 2 | *E coli* (72) | *Str. uberis* (7) |
| J (67) | 68 | 27 | 5 | *E coli* (61) | *Str. uberis* (11) |
| K (53) | 69 | 24 | 7 | *E coli* (64) | *S. aureus* (8) |
| L (53) | 48 | 44 | 8 | *E coli* (45) | *Str. uberis* (18) |
| M (53) | 70 | 28 | 2 | *E coli* (70) | NaS (12) |
| N (15) | 59 | 28 | 13 | *E coli* (55) | *Str. uberis* (8) |
| O (15) | 62 | 38 | | *E coli* (52) | *S. aureus* (17) |
| P (02) | 75 | 25 | | *E coli* (63) | other str.(13) |
| Q (39) | 45 | 49 | 6 | *E coli* (31) | *Str. uberis* (20) |
| R (64) | 74 | 18 | 8 | *E coli* (68) | *Str. uberis* (7) |
| S (74) | 68 | 30 | 2 | *E coli* (41) | *Str. uberis* (9) |

[a]The number in brackets is the code of the administrative region.

[b]*S. aureus* = *Staphylococcus aureus*, *Str. uberis* = *Streptococcus uberis*, NaS = *Non-aureus Staphylococcus*, other str. = Streptococci other than *Str. uberis* and *dysgalactiae*, and *Enterococcus*.

It is important to note that there were very few instances of "no growth" recordings (only 6.5%), which is similar to findings of some other researchers [3]. This result tends to confirm that the bacterial load in milk is high in case of severe mastitis, which makes the probability of a negative culture low. This point should be considered when treating peracute mastitis. By contrast, other studies have reported more "no growth" results, possibly due to smaller amounts of milk being tested [2,25]. In our study, we inoculated 60μL of mastitic milk (for Gram+, 30 μL per plate on COS and CNA agar, and for the majority of Gram- [Enterobacteriaceae], 30 μL per plate on COS and HEKT agar) compared to only 10μL in the other studies.

## Conclusion

Dairy producers and field veterinarians must classify the severity of clinical mastitis in cows to identify those at higher risk of developing bacteremia and needing immediate and effective therapy. Early detection is key to preventing mortality and limiting mammary functional loss. Practitioners rely on the clinical characterization of mastitis to guide their therapeutic approach. In most countries, coliforms are the primary agents isolated from severe cases of mastitis, including fatal clinical mastitis. This recent study in leading French dairy regions (Brittany, Normandy, Burgundy/Franche Comté, Grand Est region, Auvergne) identified EC as the predominant causative agent of severe clinical mastitis. This finding suggests that coliforms, especially EC, should be the main focus of first-response antibiotic therapy for severe clinical mastitis.

## Supporting information

**S1 File. Milk sample support sheet.**
(PDF)

**S2 File. Row data from veterinary clinics.**
(PDF)

**S3 File. Comparison between results obtained from veterinary clinic milk culture and obtained by MALDI TOF.**
(PDF)

## Acknowledgments

The authors wish to thank the staff of the participating dairy farms and all the French veterinary clinics for supporting the study.

## Author Contributions

**Investigation:** Florent Perrot.

**Writing – original draft:** Olivier Salat.

**Writing – review & editing:** Guillaume Lemaire, Luc Durel, Florent Perrot.

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
