## [Decision Letter · Decision Letter 0]

26 Sep 2023

PONE-D-23-23196Etiology of severe mastitis in french bovine dairy herdsPLOS ONE

Dear Dr. Salat,

Thank you for submitting your manuscript to PLOS ONE. After careful consideration, we feel that it has merit but does not fully meet PLOS ONE’s publication criteria as it currently stands. Therefore, we invite you to submit a revised version of the manuscript that addresses the points raised during the review process.

We look forward to receiving your revised manuscript.

Kind regards,

Pierre Germon

Academic Editor

PLOS ONE

3. Please include a copy of Table 2 which you refer to in your text on page 6.

Reviewers' comments:

Reviewer's Responses to Questions

**Comments to the Author**

1. Is the manuscript technically sound, and do the data support the conclusions?

Reviewer #1: Yes

Reviewer #2: Yes

2. Has the statistical analysis been performed appropriately and rigorously? 

Reviewer #1: N/A

Reviewer #2: Yes

3. Have the authors made all data underlying the findings in their manuscript fully available?

Reviewer #1: Yes

Reviewer #2: Yes

4. Is the manuscript presented in an intelligible fashion and written in standard English?

Reviewer #1: Yes

Reviewer #2: Yes

5. Review Comments to the Author

Reviewer #1: Etiology of severe mastitis in French bovine dairy herds

PONE-D-23-23196

This is a presentation of results of microbiological findings from more than 1200 severe mastitis cases in France, studied with a common method in different veterinary clinics. The study is valuable because there is little published data from France and it examined a large number of mastitis cases. The method used is adapted for practice laboratories and is therefore less specific.

I recommend language correction by a native speaker.

Inconsistent use of abbreviations. For example, in the case of E. coli, between Escherichia coli, E. coli, and EC. It is best to write out and abbreviate once the first time it is mentioned, and then use the abbreviation consistently. This is also the case for other bacteria.

L 17: Is severe mastitis really ”frequent” ? Reword please

L29: Please explain. What is the difference in severe mastitis treatment if you know the pathogen several hours later

L 54: Please find another word for: short-term lethality

L 59: Delete “ and so forth”

Ll65-67: Quinolones are also used in other areas of the world. Reword please

L 68: CM instead of MC; Why you should investigate the pathogen if you treat every severe case with quinolones?

L 83: Did you do somatic cell count testing? If not, delete “cyto-“

L 85: 30 microliters in total or per single plate?

L105-106: Is the sample considered contaminated from 1 colony of three different species or from 5 colonies of at least 3 different species?

L 108: a dot is missing after spp

L 109: It would be good if you not only describe your method, but also critically evaluate it (possibly this will come in the discussion.

L130: Table 2 is missing. Or is table 3 actually table 2?

L132: Does str. uberis account for 8-20% of the results? Table 3 also mentions 7% in some cases. See Farm I or R

L 139: The sentence is very colloquial, please reword the sentence.

L 151: catalase negative

L175ff/ Table 1: Inconsistent writing: Once comma (,) and once dots (.) are used for the percentages. In addition, when rounding to one position behind the decimal point, it is better to omit the second 0 or to remain consistent.

L 179: please explain 60 microliters or 30?

L 187: Please explain, when you mention it for the 1st time “leading French dairy regions”. What are the leading dairy regions in France?

References: Sometimes with the link from the internet and sometimes without it .

Table 2: Is there also a table 2?

Reviewer #2: In this report, Salat and colleagues report on the prevalence of the major pathogens isolated from severe clinical mastitis cases in France.

Data are clearly reported. Manuscript is well written with only a few corrections needed.

I have the following comments:

- L78: delay between call and sampling? please provide details of the sampling procedure

- L80: provide clinical report form as supplementary

- L104: specificity of the detection method? reference?

- Were samples resulting in more than 3 colony types considered contaminated samples?

- L110-117: this paragraph should be placed at the beginning of Methods section.

- How was Trueperella pyogenes identified? Could the authors provide a decision diagram?

- L130: please revise table number: table 2 is indicated in the text while the title of the table indicates Table 3.

- L131-132: please provide the number of clinics for which S. uberis was the second most frequent pathogen (11/19). Authors could also state that S. aureus is the third pathogen in prevalence (6/19).

- L145: could the authors be more precise? Are these their own data? I think these data should be presented.

- L149: period at end of sentence missing.

- L156-158: as presently written this sentence leads the reader to conclude that 60% of cases are EC while the authors are referring to Enterobacteriacae. Please rephrase.

Minor comment:

- L66: add coma after fluoroquinolones

- L68: E. coli and italicize

- In the title, although this is a short communication, I think the statement “Short communication” in the word file should be removed.

6. PLOS authors have the option to publish the peer review history of their article (what does this mean?). If published, this will include your full peer review and any attached files.

Reviewer #1: **Yes: **Volker Krömker

Reviewer #2: No

---

## [Author Response · Author response to Decision Letter 0]

29 Oct 2023

Dear reviewers, thank you very much for reviewing this article. I hope you find my changes and additions satisfactory. 

Dear editor, I've tried to comply with your recommendations

---

## [Decision Letter · Decision Letter 1]

27 Nov 2023

Etiology of severe mastitis in french bovine dairy herds

PONE-D-23-23196R1

Dear Dr. Salat,

We’re pleased to inform you that your manuscript has been judged scientifically suitable for publication and will be formally accepted for publication once it meets all outstanding technical requirements.

Kind regards,

Pierre

Academic Editor

PLOS ONE

Additional Editor Comments (optional):

Reviewers' comments:

Reviewer's Responses to Questions

**Comments to the Author**

1. If the authors have adequately addressed your comments raised in a previous round of review and you feel that this manuscript is now acceptable for publication, you may indicate that here to bypass the “Comments to the Author” section, enter your conflict of interest statement in the “Confidential to Editor” section, and submit your "Accept" recommendation.

Reviewer #1: All comments have been addressed

Reviewer #2: All comments have been addressed

2. Is the manuscript technically sound, and do the data support the conclusions?

Reviewer #1: Yes

Reviewer #2: Yes

3. Has the statistical analysis been performed appropriately and rigorously? 

Reviewer #1: N/A

Reviewer #2: Yes

4. Have the authors made all data underlying the findings in their manuscript fully available?

Reviewer #1: Yes

Reviewer #2: Yes

5. Is the manuscript presented in an intelligible fashion and written in standard English?

Reviewer #1: Yes

Reviewer #2: Yes

6. Review Comments to the Author

Reviewer #1: This is a presentation of results of microbiological findings from more than 1200 severe mastitis cases in France, studied with a common method in different veterinary clinics. The study is valuable because there is little published data from France and it examined a large number of mastitis cases. The method used is adapted for practice laboratories and is therefore less specific. The authors answered all questions and followed my suggestions. Thank you - no further comments.

Reviewer #2: Authors have adressed all the comments properly and have added the requested figures and tables. Additional inforamtion has been provided has requested.

7. PLOS authors have the option to publish the peer review history of their article (what does this mean?). If published, this will include your full peer review and any attached files.

Reviewer #1: **Yes: **Volker Krömker

Reviewer #2: No

---

## [Editor Report · Acceptance letter]

4 Dec 2023

PONE-D-23-23196R1 

Etiology of severe mastitis in French dairy herds 

Dear Dr. Salat:

I'm pleased to inform you that your manuscript has been deemed suitable for publication in PLOS ONE. Congratulations! Your manuscript is now with our production department. 

Kind regards, 

on behalf of

Dr. Pierre Germon 

Academic Editor

PLOS ONE